# Discriminant Analysis of Aroma Differences between Cow Milk Powder and Special Milk Powder (Donkey, Camel, and Horse Milk Powder) in Xinjiang Based on GC-IMS and Multivariate Statistical Methods

**DOI:** 10.3390/foods12214036

**Published:** 2023-11-05

**Authors:** Yongzhen Gou, Yaping Han, Jie Li, Xiyue Niu, Guocai Ma, Qian Xu

**Affiliations:** 1College of Food Science and Engineering, Tarim University, Alar 843300, China; 2Corps Key Laboratory of Deep Processing of Specialty Agricultural Products in Southern Xinjiang, Alar 843300, China; 3Instrumental Analysis Center, Tarim University, Alar 843300, China

**Keywords:** milk powder, finger printing, volatile compounds, GC-IMS, PLS-DA, ROC analysis

## Abstract

In order to explore the aromatic differences between Xinjiang cow milk powder and specialty milk powder (donkey, camel, and horse milk powder), Gas Chromatography-Ion Mobility Spectrometry (GC-IMS) analysis was employed to investigate the volatile compounds in these four types of milk powders. A total of 61 volatile substances were detected, with ketones, aldehydes, and alcohols being the primary flavor components in the milk powders. While the aromatic components of the different milk powders showed similarities in terms of types, there were significant differences in their concentrations, exhibiting distinct characteristics for each type. The Partial Least Squares Discriminant Analysis (PLS-DA) showed that there were 15, 14, and 23 volatile compounds that could be used for discrimination of cow milk powder against specialty milk powders, respectively. And it was validated by Receiver Operating Characteristic (ROC) analysis, and finally, 8, 6, and 19 volatile compounds were identified as valid differential marker substances. To facilitate visual discrimination between the different milk powders, we established GC-IMS fingerprint spectra based on the final discriminant markers. These studies provide theoretical guidance for the application of volatile compounds to discriminate adulteration of milk powder marketed in Xinjiang.

## 1. Introduction

Special milk refers to the milk obtained from milch animals other than cows, such as camel milk, horse milk, donkey milk, etc., which possess unique flavors and nutritional value [1]. Xinjiang, as a prominent province in animal husbandry in China, holds a significant advantage in livestock numbers and milk production [2]. In recent years, with the rapid development of the dairy industry and continuous advancements in dairy science and technology, scholars from both domestic and international domains have delved deeper into the nutritional value and physiological functions of special milk, contributing to the increasing recognition of the immense potential of specialty milks like donkey milk, horse milk, and camel milk. These specialty milks have found applications in disease adjunctive therapy, nutritional health, and food processing [3,4,5], boasting an economic value far superior to that of cow milk. However, due to the relatively low yield of specialty milks compared to cow milk and the significant impact of seasonal fluctuations on milk supply [6], along with the challenges in controlling milk source quality [7], these factors have made specialty milks and their dairy products susceptible targets for intentional adulteration driven by economic interests. As a result, incidents of adulteration motivated by financial gains occur with some frequency [8].

Milk powder serves as the most common means for preserving and transporting dairy products, carries high added value, and is widely produced and sold. Among the various fraudulent practices, the most prevalent and easily achievable one is adulterating high-value specialty milk powder with lower-value cow milk powder. This adulteration not only leads to economic losses for consumers but also poses potential health risks, especially for individuals allergic to cow milk [9]. Unfortunately, there are currently no clear standards for detecting adulteration in specialty milk powders [10], and market supervision lacks reliable criteria and testing methods. Furthermore, there is a paucity of research concerning the detection of adulteration in specialty milk powders. Thus, there is an urgent need to address the issue of adulteration in Xinjiang’s specialty milk powders by establishing screening, analytical testing methods, and corresponding standards to safeguard the interests of general consumers.

Currently, the most common technique used to discriminate between different milk powder sources is based on DNA sequencing. Species-specific DNA sequences have been widely applied in polymerase chain reaction (PCR) detection, enabling the identification of animal milk sources and thereby detecting adulteration in milk [11,12]. However, PCR methods involve intricate DNA separation procedures and typically do not yield quantitative information. Specialty milk powders and cow milk powders share similarities in appearance and major chemical components, but their physiological metabolic lactation activities vary among different animal species [13]. Additionally, their milk powder processing methods also differ, wherein volatile compounds can serve as a basis for distinguishing specialty milk powders from cow milk powder. Compared to DNA sequencing-based methods, studying the differences in volatile compounds between cow milk powder and specialty milk powder proves more practical, convenient, and rapid. Commonly employed techniques include Gas Chromatography-Mass Spectrometry (GC-MS) and Gas Chromatography-Ion Mobility Spectrometry (GC-IMS).

GC-IMS is an emerging detection technique in food aroma analysis, known for its high sensitivity, strong convenience, short analysis time, and data visualization advantages, making it widely applied in food quality control and traceability [14]. Gomez et al. [15] non-destructively analyzed volatile compounds in spoiled and unspoiled ham using GC-IMS. They established a volatile compound model to classify the freshness of ham through PLS-DA. He et al. [16] used a combination of Gas Chromatography-Mass Spectrometry (GC-MS) and GC-IMS to analyze volatile compounds during different distillation stages of white spirit. By employing PLS-DA, they identified four aroma-active markers and validated them using ROC analysis. Ethyl butyrate and ethyl hexanoate were ultimately determined as the most effective discriminant markers for distinguishing different distillation stages. Feng et al. [17] compared the influence of two drying methods (spray drying and freeze drying) on the volatile organic compounds of yak milk powder using GC-IMS and Principal Component Analysis (PCA). They established characteristic fingerprint spectra for yak milk powder under different drying methods.

The literature suggests that the application of GC-IMS in milk powder analysis is feasible and is still in its preliminary stage. There are scarce reports on the use of GC-IMS for discriminating between different types of milk powders. Therefore, this study innovatively employs GC-IMS to analyze and investigate the variations in volatile compounds between cow milk powder and specialty milk powder. The research aims to utilize PLS-DA to identify volatile compounds that can effectively discriminate between cow milk powder and specialty milk powder. The selected volatile compounds will undergo validation and screening through ROC analysis. Subsequently, the final chosen volatile compounds will be used to establish a rapid identification fingerprint spectrum and provide a theoretical basis for the application of volatile compounds to discriminate against the adulteration of commercially available milk powder in Xinjiang. 

## 2. Materials and Methods 

### 2.1. Preparation and Collection of Milk Powder Samples

Samples of cow milk, camel milk, horse milk, and donkey milk were collected by 2022 Xinnong Enterprises (Aral, China) in February, June, and October on behalf of the three batches of collection, and three samples were collected in each batch, and the collected milk samples were then transported to the laboratory at −80 °C in a liquid nitrogen tank to be preserved and stored at −80 °C, and experiments were carried out after all the samples were collected. 

In the laboratory of Xinnong Enterprises (Aral, China), experiments were carried out to produce milk powder. This aimed to simulate the market products of milk powder production companies in Xinjiang. Cow and camel milk powder were produced using spray drying, while horse and donkey milk powders were produced through freeze-drying on the market. Drying experiments were conducted on different milk types, as detailed in Table 1.

Spray drying: The spray drying equipment ( Oumeng Laboratory specialized small spray dryer OM-1500A, Shanghai, China) operated with the following parameters: inlet air temperature of 120 °C, an outlet air temperature of 65 °C, and a flow rate of 5 mL/min. The milk powder samples were spray-dried until their moisture content reached 5–6%.

Freeze drying: Firstly, the milk samples were pre-frozen and then placed in a −20 °C freezer for 6 h until they were completely frozen. After the initial freezing, the samples were transferred to a freeze dryer (Jinan Junde FD-503 box-type vacuum freeze dryer, Jinan, China) with the following settings: vacuum at 20 Pa, sublimation temperature at 40 °C, and freeze trap temperature at −40 °C. The milk powder’s moisture content was reduced to 5–6%. Before analysis, each sample was stored in dry containers.

### 2.2. GC-IMS Analysis

2.0 g of sample was placed into a 20 mL headspace vial (Agilent, Santa Clara, CA, USA) and incubated at 80 °C for 20 min while stirring at 500 rpm. After the incubation was completed, the temperature of the injection needle was 85 °C, and 500 μL of gas from the headspace vial was automatically aspirated into the GC-IMS instrument (FlavorSpec, GAS, Dortmund, Germany) without shunting. The samples were subjected to compound separation using a WAT—Wax capillary column (30 m × 0.53 mm, 1 μm) at a constant temperature of 60 °C, thereby standardizing the entire experimental procedure. The flow rate was controlled as follows: initially held at a constant 2 mL/min for 2 min, increased to 10 mL/min over 8 min, and then increased to 100 mL/min over 10 min, held for 10 min. The drift tube was 98.0 mm long, the operating temperature was 45 °C, and the constant voltage was 5 kV. The entire analysis took 45 min. All analyses were repeated three times. The obtained GC-IMS data were observed using Laboratory Analytical Viewer (G.A.S, Dortmund, Germany). The Reporter plug-in was used to directly compare the 2D top view and difference plots. The retention indices (RI) of volatile compounds were calculated using ortho-ketones C4~C9 (Sinopharm Chemical Reagent Co., Ltd., Beijing, China) as external standards. The volatile compounds were identified by comparing the retention index (RI) with the normalized drift time (RIP) in the IMS database (G.A.S., Dortmund, Germany). Fingerprints were made and compared using the Gallery Plot plugin.

### 2.3. Statistical Analysis

The key differential components between different milk powders were explored using SIMCA-P 13.0 software (Umetrics, Stockholm, Sweden) with Partial Least Squares Discriminant Analysis (PLS-DA). To evaluate the differences in volatile compounds among the various milk powders, One-Way Analysis of Variance (ANOVA) was conducted using SPSS Statistics 20.0 software (SPSS Inc., Chicago, IL, USA). The significance level was set at *p* < 0.05. The identification and validation of discriminant volatile compounds were carried out through ROC analysis.

## 3. Results and Analysis

### 3.1. GC-IMS Profiling of Different Milk Powders

Currently, GC-MS is one of the most commonly used techniques for analyzing volatile compounds. However, the requirement for a vacuum environment and its bulky size limit its portability [18]. On the other hand, GC-IMS, as an emerging instrument analysis technology, effectively separates ions based on their migration velocity at atmospheric pressure. It offers advantages such as high sensitivity, low environmental requirements, portability, and data visualization. Consequently, GC-IMS finds widespread application in food classification and quality control [19].

In this study, gas chromatography-ion mobility spectrometry (GC-IMS) analysis was conducted using the FlavorSpec instrument to generate data, as shown in Figure 1a. The figure presents a 2D topographical map obtained from a 3D spectrum scan, comparing the volatile compounds in different milk powders. The X-axis represents ion migration time, and the Y-axis represents the retention time of ions in gas chromatography. To eliminate the influence of capillary column temperature and carrier gas flow rate, the response ion peak at X = 1.0 was normalized, as indicated by the red vertical line in the plot. The color represents the signal intensity of individual compounds, with red indicating a higher concentration and blue indicating a lower concentration. Most of the signals are distributed within the retention time range of 100–1600 s, and many peaks show similar distributions within the samples, but with varying intensities among the same type of samples. Additionally, there are evident differences in peak signals among different samples, indicating variations in the content of volatile compounds in different milk powders. Particularly, there are distinct differences in the drying methods, with samples D and H (freeze-dried) having more peaks compared to samples M and C (spray-dried). This discrepancy may be attributed to the production of some volatile substances during the vacuum freezing process [20], or it could be due to the pre-freezing of water before drying, potentially reducing the diffusion of surface volatile compounds. Spray drying may also cause changes and the loss of some volatile compounds due to the high temperature conditions [21].

To clearly compare the differences between samples from different milk powders, this study utilized a difference comparison mode, as depicted in Figure 1b. The ion mobility chromatogram of sample M-1 was chosen as the reference, and the ion mobility chromatograms of other samples were subtracted from the reference. If two volatile compounds are the same, the subtracted background would be white. Red dots indicate that the concentration of a compound is higher than the reference concentration, while blue dots indicate that the concentration is lower. Comparing samples M-2 and M-3 to other different milk powder samples, the areas of red and blue are significantly reduced. This suggests that the volatile compound content of the reference sample M-1 is similar to samples M-2 and M-3 but significantly differs from other samples. By establishing difference comparison plots for different milk powders using gas chromatography-ion mobility spectrometry, preliminary and simple discrimination between different milk powders can be achieved.

### 3.2. Analysis of Volatile Matter Content Based on GC-IMS

In order to study the characteristics of volatile compounds in different milk powders, GC-IMS was employed to analyze and identify the volatile compound components and their relative concentrations in four different samples. As depicted in Figure 1, a total of 108 peak signals were detected. Despite GC-IMS’s high sensitivity, due to the limitations of the database, only 61 compounds could be qualitatively identified. Among these identified compounds, there were 22 dimers, where water and hydrogen ions have the possibility of combining with one monomer and two monomers to form a monomer and a dimer when charged volatile compounds are given [22]. This enhances the accuracy of GC-IMS for the quantification of volatile compounds. Analyzing the types of identified compounds, Table 2 shows the 61 volatile flavor compounds identified by GC-IMS, including 13 alcohols, 7 esters, 12 aldehydes, 7 ketones, 1 ether, 1 acid, and 2 furan compounds. Compared to previous reports [23], GC-IMS demonstrated higher sensitivity to aldehydes, ketones, and alcohols than to alkanes, which is consistent with the results reported by Gou [24]. The aroma components of the four milk powder samples were similar in type, but their total concentrations varied. Aldehydes and ketones had the highest content, followed by alcohols and esters. Among them, aldehydes and ketones accounted for a significant proportion of the overall composition, making them important volatile compounds in milk powders. This aligns with the findings of Bai et al. [25] in their analysis of donkey milk powder using GC-MS, as well as with the results of Chi et al. [26] in their examination of bovine milk powder. It is worth noting that GC-IMS did not detect a significant proportion of hydrocarbon compounds, owing to the fundamental principles of GC-IMS analysis [27], which exhibit limited sensitivity towards hydrocarbons.

Ketones are common flavor compounds in milk powder, often present in the form of methyl ketones [28]. They have a low threshold value and play an extremely important role in the overall flavor of milk powder [29]. During the processing and storage of milk powder, free fatty acids in the sample are first oxidized to 8-ketonic acids, which are then decarboxylated into corresponding methyl ketones [30]. As shown in Table 2, the detected ketones in the four milk powders accounted for 23.41%, 10.83%, 15.46%, and 9.69% of the total volatile compounds, respectively. Among them, acetone, 3-hydroxy-2-butanone, 2-butanone, and 2-heptanone were relatively abundant. 3-Hydroxy-2-butanone imparts a milky aroma and is a representative volatile flavor compound in dairy products [20]. 2-heptanone and 2-butanone are associated with fruity, sweet, and slightly milky aromas [31]. In terms of differences between different milk powders, the total ketone content of sample M and sample C was significantly higher than that of sample D and sample H. This could be due to the conditions of spray drying, which may disrupt the coalescence of milk droplets, releasing free fatty acids that support oxidative reactions and increasing the total ketone content [32]. As for the content of 1-penten-3-one, sample M and sample C had significantly lower levels than sample D and sample H. This discrepancy may be attributed to the differences in the content and types of free fatty acids in different types of milk [33].

Aldehydes are one of the main flavor compounds in milk powder. The generation of aliphatic and enal aldehydes with more than 5 carbon atoms in milk powder mainly occurs through the oxidation of higher homologous fatty acids [34]. For example, in this study, aldehydes such as 1-hexanal, 1-nonanal, heptanal, and 1-octanal were detected. According to Li et al. [35], these substances can be used as indicators to evaluate the oxidative flavor of milk powder. Aldehydes with less than six carbon atoms are produced by PUFA oxidation [36]. Aldehydes in milk powder have a lower overall content than ketones but have a lower threshold and a significant impact on the overall flavor of the milk powder. Low concentrations of aliphatic aldehydes typically exhibit herbal aromas, giving the milk powder a fresh taste, while high concentrations may lead to undesirable odors [37]. Among the four milk powders, the aliphatic aldehydes accounted for 16.65%, 22.21%, 21.00%, and 23.81% of the total volatile compounds, respectively. The total aldehyde content of sample M was significantly lower than that of sample D, sample C, and sample H. This difference could be attributed to variations in the content of free fatty acids and free amino acids in different liquid milk types.

Alcohol compounds have a higher threshold and contribute to the flavor of milk only when they are present in an unsaturated state or at high concentrations [38], but they are still important components of milk powder flavor. Alcohol compounds in milk powder may be the result of the reduction reaction of corresponding aldehyde compounds [39], and their content is positively correlated with the content of aldehyde compounds. 4-terpenol is likely to be synthesized by plants and some microorganisms and transferred from feed to milk through the rumen [40], so its content is unrelated to aliphatic aldehydes in milk powder but rather related to the forage fed to the livestock.

Ester compounds, due to their low threshold value, are essential flavor compounds in milk powder, imparting buttery, fruity, and floral aromas and helping to reduce the off-flavors of fatty acids and miscellaneous alcohols in milk powder [41]. Ester compounds are mainly formed by the esterification reaction between free fatty acids and alcohols present in milk fat. Some ester compounds are produced by bacterial esterases, and their detection can indicate the degree of bacterial contamination during the raw material processing and storage of milk powder. Dimethyl sulfide is the only sulfur compound detected by GC-IMS. It may be through the Strecker reaction involving riboflavin and methionine [42]. Low concentrations of dimethyl sulfide contribute to the ideal flavor of fresh milk and milk powder, but at high concentrations, it can cause undesirable off-flavors. The detected furan derivatives include 2-pentylfuran and 2-ethylfuran, which result from the Maillard reaction and oxidation of lactose and amino acids [43]. They have nutty and vanilla-like flavors [44].

However, merely observing the content of volatile compounds in various milk powders makes it challenging to classify them effectively. Therefore, the volatile compound content of different milk powders is utilized as feature data for subsequent PLS-DA statistical multivariate analysis.

### 3.3. Analysis of the Differences between Cow Milk Powder and Other Kinds of Milk Powder Based on PLS-DA

PLS-DA is a supervised discriminant analysis statistical method used to establish a relationship model between volatile compounds and sample categories, enabling the prediction of sample categories [18]. In this study, a PLS-DA model was employed to identify specific marker compounds in cow milk powder samples that differentiate them from the other three samples based on the detected volatile compound content data matrix. The results are shown in Figure 2, where the contributions of Partial Least Squares Discriminant Analysis (PLS-DA) for the three score plots are 64%, 56.8%, and 60.4%, respectively. Contributions exceeding 50% indicate successful differentiation between these three groups of analyses. The samples are stably distributed in the first and fourth quadrants and the second and third quadrants. Additionally, a permutation test was conducted on these three models, and the model prediction index (Q2) for each was found to be 0.67, 0.70, and 0.60, respectively, with prediction rates of 67%, 70%, and 60% for distinguishing different milk powders. A Q2 value exceeding 0.5 indicates an acceptable model fit, and the Q2 regression line intersecting the y-axis below zero suggests no overfitting, making the model validation reliable [22]. Therefore, these results can be used for milk powder identification analysis.

To determine the most statistically significant variables, the Variable Importance in the Projection (VIP) value variable selection method was applied. Generally, volatile compounds with VIP values > 1.0 are considered to play an important role in discrimination [45]. In Figure 2a, the red markers indicate a total of 15 substances, namely 2-Nonanone, (Z)-2-pentenol, ethyl butyrate, (Z)-4-heptenal, (E, E)-2,4-heptadienal, 1-hexanal, (E)-2-Pentenal, 2-Pentanone, acetone, dimethyl sulfide, 2-Butanone, 4-Terpinenol, 1-Penten-3-ol, 3-Hydroxy-2-butanone, and 2-heptanone, as the marker volatile compounds distinguishing cow milk powder from donkey milk powder. In Figure 2b, the red markers indicate a total of 14 substances, namely (E)-2-Heptenal, 4-Terpinenol, 1-nonanal, (E)-2-Pentenal, 3-methyl-1-Butanol, (Z)-2-pentenol, Heptanal, acetone, 3-Hydroxy-2-butanone, 2-Pentanone, 2-methyl butanal, 1-Penten-3-one, propyl acetate, and 2-heptanone, as the marker volatile compounds distinguishing cow milk powder from camel milk powder. In Figure 2c, the red markers indicate a total of 23 substances, namely 1-Penten-3-ol, (E,E)-2,4-heptadienal, 2-Nonanone, 1-octanal, 2-Methyl-1-propanol, 2-ethylfuran, 2-methyl butanal, 3-methyl-1-Butanol, Ethyl propanoate, Propanal, dimethyl sulfide, (E)-2-hexenal, 2-Butanone, 1-hexanol, 3-Hydroxy-2-butanone, 4-Terpinenol, (Z)-4-heptenal, 2-heptanone, (Z)-2-pentenol, acetone, (E)-2-Pentenal, and (Z)-2-Methylpent-2-enal, as the marker volatile compounds distinguishing cow milk powder from horse milk powder.

However, the models derived from the PLS-DA analysis have not undergone precise validation. Therefore, it is imperative to employ ROC analysis for more accurate model calibration.

### 3.4. Validation of PLS-DA Screening Substances Based on Roc Analysis and Yoden Index Discrimination

The ROC curve is a convenient and efficient tool used in the study of binary classification problems. It plots the true positive rate against the false positive rate at different cutoff points or thresholds, constructing a monotonically increasing curve. The area under the ROC curve (AUC) serves as a measure of discriminatory performance, where a larger AUC value indicates a more effective classification method [46]. In order to assess the discriminative ability of the key volatile compounds identified by PLSD-DA between cow milk powder and the other three types of milk powders, ROC curve analysis was conducted using SPSS software. The Jordon index of these volatile compounds is also calculated, and the Jordon index maximum value selection is performed to determine the maximum threshold for the discrimination of the substance.

In the application of the ROC curve analysis, by considering both sensitivity and specificity results at various cutoff points [47], we selected the volatile compounds with AUC values between 0.95 and 1. These compounds have good discriminatory ability for different types of milk powder and are more representative of distinguishing milk powders. For cow milk powder versus donkey milk powder discrimination, the final discriminatory markers were determined as (E)-2-Pentenal, 2-Pentanone, acetone, dimethyl sulfide, 2-Butanone, 4-Terpinenol, 3-Hydroxy-2-butanone, and 2-heptanone, as shown in Figure 3a. For cow milk powder versus camel milk powder discrimination, the final discriminatory markers were determined as 3-methyl-1-butanol, 3-hydroxy-2-butanone, 2-methyl butanal, 1-penten-3-one, propyl acetate, and 2-heptanone, as shown in Figure 3b. For cow milk powder versus horse milk powder discrimination, the final discriminatory markers were determined as 2-nonanone, 2-methyl-1-propanol, 2-methyl butanal, 3-methyl-1-Butanol, ethyl propanoate, Propanal, dimethyl sulfide, (E)-2-hexenal, 2-Butanone, 1-hexanol, 3-hydroxy-2-butanone, 4-Terpinenol, (Z)-4-heptenal, 2-heptanone, (Z)-2-pentenol, acetone, (E)-2-pentenal, 1-penten-3-ol, and (Z)-2-methylpent-2-enal, as shown in Figure 3c. The thresholds determined by the Youden index in Figure 3 also indicate the maximum values for each compound during the corresponding discrimination process, making the classification more accurate.

Among these, (E)-2-pentenal and 4-terpinenol are downregulated volatile compounds in both cow milk powder and donkey milk powder, and they are also downregulated in the comparison between cow and horse milk powder. 3-hydroxy-2-butanone, 2-heptanone, and ketone-like substances are upregulated in the comparison between cow and specialty milk powder. This phenomenon is attributed to the intestinal microbiota, which breaks down plant materials and transfers them into milk [48]. It may be due to differences in the food breakdown process between cows and donkeys. Cows and camels, being ruminants, primarily digest and absorb nutrients in their four stomach compartments, whereas for horses and donkeys, which lack a complex multi-chambered stomach, the primary site for digesting and absorbing nutrients is in the intestines [49]. These distinct species-specific processes result in variations in the composition of the produced milk, with the most significant impact on volatile compounds being the differences in homologous fatty acids and unsaturated fatty acids. Lu et al. [33] found significant differences in the types of fatty acids in different milk powders, especially between cow and horse milk powder. Dimethyl sulfide, a substance produced in large quantities through a high-temperature oxidation [50], is upregulated in the comparison between cow and horse milk powder but is not shown in the comparison with camel milk powder.

However, the discriminative substances obtained are not represented specifically with numerical values. Therefore, the decision was made to employ GC-IMS fingerprint spectra to showcase the discriminative substances.

### 3.5. Validation of the Production of GC-IMS Discriminant Fingerprints of Cow Milk Powder against Other Milk Powders Based on ROC Analysis

For the expeditious and convenient discrimination of cow milk powder from other milk powders, the GC-IMS fingerprint spectra of volatile compounds validated through ROC analysis were established, as illustrated in Figure 4. The fingerprint spectra were created by assembling cutouts of volatile compounds from Figure 1a. Each row represents a sample, and each column represents a specific compound, with brighter and redder shades indicating higher concentrations, while darker and bluer shades suggest lower concentrations. By comparing the textual and quantitative attributes of volatile compounds, the GC-IMS fingerprint spectra enable to clearly assess the varying levels of volatile compounds among different milk powders and discern distinctive characteristics in their volatile profiles. In Figure 4a, the distinctive features of volatile compounds such as 2-butanone facilitate rapid and precise discrimination between cow and donkey milk powder. In Figure 4b, the distinctive features of volatile compounds like 3-hydroxy-2-butanone allow for clear and swift discrimination between cow and camel milk powder. Similarly, in Figure 4c, the distinctive features of volatile compounds including dimethyl sulfide enable a clear and rapid discrimination between cow and camel milk powder.

## 4. Conclusions

Using GC-IMS, the volatile compounds of cow milk powder, donkey milk powder, horse milk powder, and camel milk powder were detected, resulting in a total of 61 volatile compounds. Ketones, aldehydes, and alcohols were found to be the primary flavor components in the milk powders. While the aroma compositions of different milk powders were similar in type, significant differences were observed in their concentrations, exhibiting distinct characteristics for each type. GC-IMS demonstrated a higher sensitivity to aldehydes and ketones compared to alkanes, based on the preference in detecting various substances. Through multivariate statistical analysis (PLS-DA and ROC validation), specific volatile compounds were identified that can discriminate between different types of milk powders and the maximum threshold for discrimination was established. For cow and donkey milk powder, (E)-2-pentenal and other eight volatile compounds were identified as the ultimate discriminatory markers. For cow and camel milk powder, 3-methyl-1-butanol and other six volatile compounds served as the final discriminatory markers. For cow and horse milk powder, 2-nonanone and other 19 volatile compounds were determined as the ultimate discriminatory markers. These findings provide a theoretical basis for detecting adulterated cow milk powder in special milk powders. To facilitate the swift and convenient analysis of research results, a fingerprint spectrum was established for distinguishing cow milk powder from different types of milk powders. It is essential to emphasize that the application of GC-IMS enables the identification of different food products, ensuring the authenticity of food and detecting potential adulteration, especially for food items with complex testing methods. 

## Figures and Tables

**Figure 1 foods-12-04036-f001:**
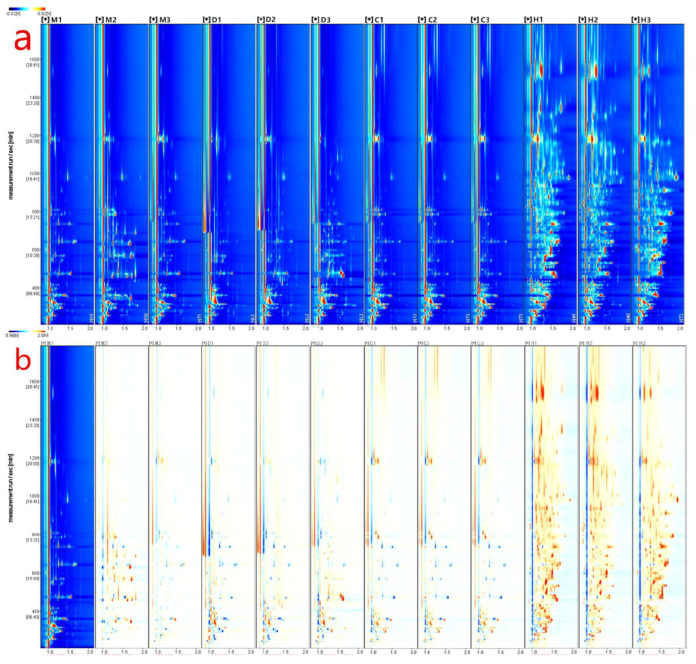
GC−IMS analysis of different milk powders. (**a**) 3D Spectrogram of volatile components in different milk powders−top view; (**b**) 2D Comparative chart of volatile components in different milk powders. Wherein, M, D, C, and H denote bovine milk powder, donkey milk powder, camel milk powder, and equine milk powder, respectively.

**Figure 2 foods-12-04036-f002:**
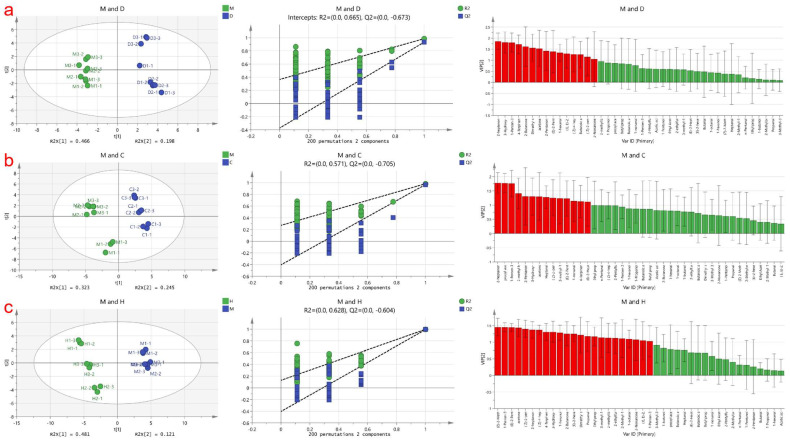
Presents the PLS−DA analysis based on cow milk powder samples compared to the other three types of milk powder samples. (**a**) Displays the score plot, cross-validation results obtained from 200 permutation tests, and the plot of volatile compounds arranged according to VIP scores for discriminating between cow milk powder and donkey milk powder; (**b**) Represents the PLS−DA analysis plot for discriminating between cow milk powder and camel milk powder; (**c**) Illustrates the PLS−DA analysis plot for discriminating between cow milk powder and horse milk powder.Where the red part is VIP > 1 and the green part is VIP < 1.

**Figure 3 foods-12-04036-f003:**
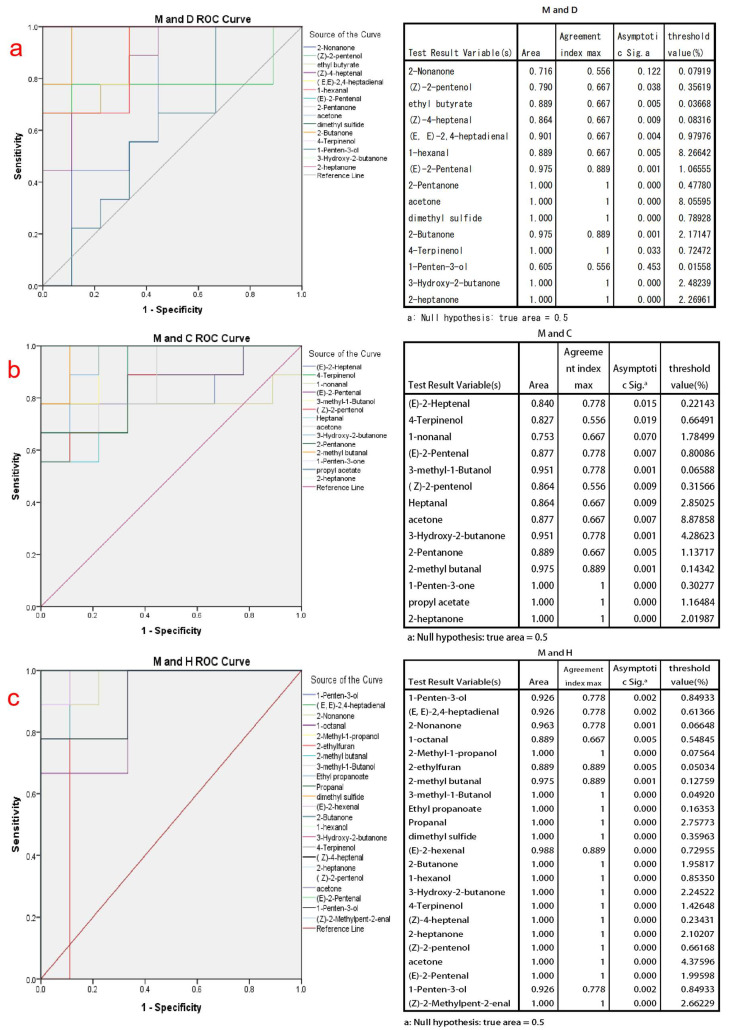
The ROC analysis graphs and the maximum Youden index threshold discrimination table are based on the initial screening of discriminative compounds using PLS-DA analysis for cow milk powder samples compared to the other three types of milk powder samples. (**a**) The ROC analysis validation graph for discriminating between cow milk powder and donkey milk powder; (**b**) The ROC analysis validation graph for discriminating between cow milk powder and camel milk powder; (**c**) The ROC analysis validation graph for discriminating between cow milk powder and horse milk powder. Note: The lines with colors not appearing in the ROC analysis graphs coincide with the upper-left boundary.

**Figure 4 foods-12-04036-f004:**
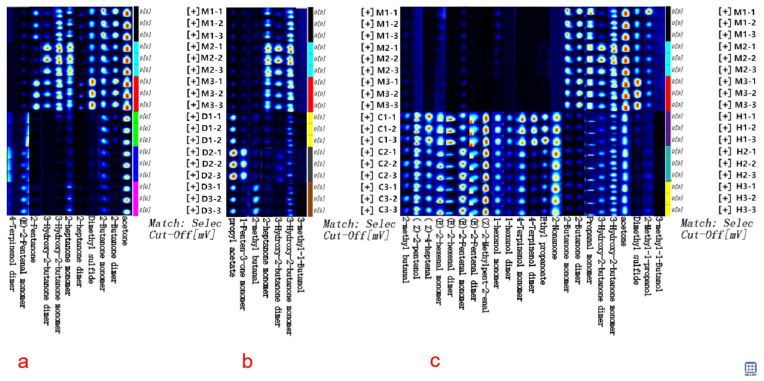
The GC-IMS fingerprint spectra are based on ROC validation for discriminating volatile compounds. (**a**) The fingerprint spectra for discriminating between cow and donkey milk powder; (**b**) The fingerprint spectra for discriminating between cow and camel milk powder; (**c**) The fingerprint spectra for discriminating between cow and horse milk powder.

**Table 1 foods-12-04036-t001:** Information on different milk powder samples.

Description of Sample	Abbreviation	Sampling Site	Processing Methods
Cow milk	M	Aksu Xinjiang	Spray drying
Donkey milk	D	Hetian Xinjiang	Freeze drying
Camel milk	C	Hetian Xinjiang	Spray drying
Horse milk	H	Altay Xinjiang	Freeze drying

**Table 2 foods-12-04036-t002:** Volatile matter content of different milk powders.

Compound	Formula	MW	RI	Dt	Contents (%)			
					M	D	C	H
Ketones					23.41 ± 4.63 ^a^	10.83 ± 2.46 ^c^	15.46 ± 2.27 ^b^	9.69 ± 1.24 ^c^
Acetone	C_3_H_6_O	58.1	809.6	1.11185	9.28 ± 0.85 ^a^	6.80 ± 1.24 ^c^	8.06 ± 0.50 ^b^	2.86 ± 0.07 ^d^
3-Hydroxy-2-butanone *	C_4_H_8_O_2_	88.1	1285.1	1.0663	5.60 ± 1.58 ^a^	0.55 ± 0.14 ^c^	3.16 ± 0.31 ^b^	1.41 ± 0.31 ^c^
2-Butanone *	C_4_H_8_O	72.1	898.1	1.06083	3.10 ± 0.72 ^a^	1.30 ± 0.59 ^b^	2.62 ± 0.44 ^a^	1.21 ± 0.41 ^b^
2-Heptanone *	C_7_H_14_O	114.2	1172.8	1.26016	3.51 ± 0.52 ^a^	1.20 ± 0.23 ^c^	0.78 ± 0.20 ^d^	1.51 ± 0.06 ^b^
2-Pentanone	C_5_H_10_O	86.1	973.4	1.36654	1.61 ± 0.93 ^a^	0.29 ± 0.13 ^b^	0.42 ± 0.15 ^b^	1.38 ± 0.21 ^a^
1-Penten-3-one *	C_5_H_8_O	84.1	1018.2	1.0789	0.24 ± 0.02 ^d^	0.61 ± 0.12 ^b^	0.34 ± 0.01 ^c^	1.39 ± 0.07 ^a^
2-Nonanone	C_9_H_18_O	142.2	1394.8	1.87621	0.06 ± 0.009 ^b^	0.08 ± 0.003 ^b^	0.06 ± 0.01 ^b^	0.20 ± 0.11 ^a^
Aldehydes					16.65 ± 6.96 ^b^	22.21 ± 7.96 ^a^	21.00 ± 6.24 ^a^	23.81 ± 2.88 ^a^
1-Hexanal *	C_6_H_12_O	100.2	1095.9	1.26878	4.38 ± 2.78 ^b^	9.02 ± 2.47 ^a^	6.19 ± 2.51 ^b^	4.69 ± 0.07 ^b^
Propanal *	C_3_H_6_O	58.1	773.8	1.05825	3.996 ± 0.94 ^b^	3.86 ± 1.16 ^b^	4.27 ± 1.00 ^a^	2.18 ± 0.24 ^b^
1-Nonanal *	C_9_H_18_O	142.2	1408.2	1.47784	1.93 ± 0.63 ^a^	1.43 ± 0.64 ^ab^	1.26 ± 0.14 ^b^	1.53 ± 0.45 ^ab^
Heptanal *	C_7_H_14_O	114.2	1173.6	1.33622	1.77 ± 0.70 ^bc^	2.10 ± 1.19 ^b^	3.35 ± 1.02 ^a^	1.19 ± 0.17 ^c^
n-Pentanal	C_5_H_10_O	86.1	975.7	1.42324	0.95 ± 0.39 ^b^	0.86 ± 0.07 ^b^	1.53 ± 0.64 ^a^	0.98 ± 0.18 ^b^
(E)-2-Hexenal *	C_6_H_10_O	98.1	1207.7	1.17953	0.44 ± 0.23 ^b^	0.52 ± 0.09 ^b^	0.46 ± 0.07 ^b^	1.31 ± 0.31 ^a^
(E)-2-Pentenal *	C_5_H_8_O	84.1	1129.4	1.1047	0.75 ± 0.15 ^c^	1.30 ± 0.39 ^b^	0.95 ± 0.08 ^c^	3.19 ± 0.16 ^a^
Butanal	C_4_H_8_O	72.1	898.5	1.28609	1.12 ± 0.69 ^a^	0.87 ± 0.25 ^a^	1.29 ± 0.45 ^a^	1.09 ± 0.27 ^a^
1-Octanal *	C_8_H_16_O	128.2	1283.7	1.40792	0.50 ± 0.16 ^bc^	0.44 ± 0.06 ^b^	0.59 ± 0.09 ^b^	0.86 ± 0.19 ^a^
(E,E)-2,4-Heptadienal *	C_7_H_10_O	110.2	1541	1.19486	0.54 ± 0.19 ^b^	1.00 ± 0.34 ^a^	0.59 ± 0.03 ^b^	1.30 ± 0.55 ^a^
(Z)-2-Methylpent-2-enal	C_6_H_10_O	98.1	1153.5	1.49675	0.22 ± 0.008 ^b^	0.24 ± 0.03 ^b^	0.19 ± 0.01 ^b^	5.34 ± 0.25 ^a^
2-Methyl butanal	C_5_H_10_O	86.1	916.1	1.40205	0.10 ± 0.02 ^b^	0.55 ± 0.06 ^a^	0.36 ± 0.01 ^ab^	0.14 ± 0.01 ^b^
Alcohols					9.84 ± 2.87 ^a^	8.83 ± 1.79 ^a^	9.85 ± 2.82 ^a^	13.10 ± 1.95 ^b^
1-Pentanol *	C_5_H_12_O	88.1	1247.7	1.25373	2.95 ± 0.18 ^ab^	1.92 ± 1.60 ^a^	2.92 ± 1.13 ^ab^	3.48 ± 0.20 ^b^
1-Butanol *	C_4_H_10_O	74.1	1136.8	1.18259	2.41 ± 0.13 ^a^	2.27 ± 0.73 ^a^	1.61 ± 0.59 ^ab^	1.15 ± 0.02 ^b^
1-Propanol *	C_3_H_8_O	60.1	1026.8	1.11126	1.11 ± 0.63 ^b^	0.67 ± 0.16 ^d^	1.58 ± 0.28 ^a^	1.05 ± 0.05 ^c^
1-Penten-3-ol	C_5_H_10_O	86.1	1154.6	0.94065	1.03 ± 0.28 ^a^	1.28 ± 0.49 ^a^	1.32 ± 0.32 ^a^	0.65 ± 0.12 ^b^
1-Hexanol *	C_6_H_14_O	102.2	1366.7	1.32911	0.48 ± 0.22 ^c^	0.41 ± 0.07 ^c^	0.70 ± 0.24 ^b^	1.07 ± 0.07 ^a^
4-Terpinenol *	C_10_H_18_O	154.3	1590.2	1.22368	0.58 ± 0.08 ^b^	0.95 ± 0.13 ^b^	0.68 ± 0.06 ^b^	3.63 ± 1.12 ^a^
2-Methyl-1-propanol	C_4_H_10_O	74.1	1089.8	1.17393	0.26 ± 0.15 ^a^	0.32 ± 0.01 ^a^	0.28 ± 0.06 ^a^	0.03 ± 0.004 ^b^
(Z)-2-Pentenol	C_5_H_10_O	86.1	1315.9	0.94353	0.42 ± 0.08 ^b^	0.33 ± 0.03 ^c^	0.32 ± 0.04 ^c^	0.82 ± 0.01 ^a^
(E)-2-Heptenal	C_7_H_12_O	112.2	1321.8	1.2579	0.29 ± 0.10 ^b^	0.25 ± 0.05 ^bc^	0.18 ± 0.02 ^c^	0.39 ± 0.09 ^a^
3-Methyl-3-buten-1-ol	C_5_H_10_O	86.1	1254.8	1.17277	0.12 ± 0.005 ^b^	0.12 ± 0.001 ^b^	0.10 ± 0.02 ^b^	0.21 ± 0.08 ^a^
3-Methyl-1-butanol	C_5_H_12_O	88.1	1196.5	1.24773	0.13 ± 0.06 ^ab^	0.23 ± 0.02 ^a^	0.057 ± 0.01 ^b^	0.03 ± 0.003 ^b^
(Z)-4-heptenal *	C_7_H_12_O	112.2	1232.8	1.14759	0.064 ± 0.001 ^b^	0.10 ± 0.003 ^b^	0.10 ± 0.03 ^b^	0.6 ± 0.14 ^a^
Esters					9.45 ± 2.88 ^a^	8.17 ± 2.04 ^b^	9.03 ± 2.25 ^a^	6.94 ± 2.37 ^c^
Ethyl acetate	C_4_H_8_O_2_	88.1	869.4	1.33873	4.3 ± 2.44 ^ab^	2.66 ± 3.31 ^b^	5.05 ± 1.60 ^a^	3.15 ± 0.27 ^ab^
Acetic acid butyl ester *	C_6_H_12_O_2_	116.2	1065.2	1.23989	2.3 ± 0.24 ^ab^	4.69 ± 2.8 ^ab^	0.78 ± 0.009 ^b^	2.19 ± 1.24 ^a^
Propyl acetate	C_5_H_10_O_2_	102.1	969.8	1.47649	0.57 ± 0.11 ^c^	0.34 ± 0.03 ^c^	2.93 ± 0.50 ^a^	1.03 ± 0.64 ^b^
Butyl propanoate	C_7_H_14_O_2_	130.2	1134.9	1.71778	1.67 ± 2.28 ^a^	0.20 ± 0.03 ^b^	0.14 ± 0.02 ^b^	0.07 ± 0.004 ^b^
Butanoic acid, butyl ester	C_8_H_16_O_2_	144.2	1203.1	1.8173	0.44 ± 0.05 ^a^	0.09 ± 0.02 ^b^	0.06 ± 0.008 ^b^	0.04 ± 0.002 ^b^
Ethyl propanoate	C_5_H_10_O_2_	102.1	952.7	1.45226	0.089 ± 0.05 ^b^	0.082 ± 0.03 ^b^	0.046 ± 0.006 ^b^	0.37 ± 0.15 ^a^
Butanoic acid ethyl ester	C_6_H_12_O_2_	116.2	1032.3	1.55874	0.048 ± 0.002 ^b^	0.10 ± 0.04 ^a^	0.032 ± 0.004 ^b^	0.09 ± 0.006 ^a^
Other					6.07 ± 1.87 ^a^	4.92 ± 1.54 ^b^	2.71 ± 0.76 ^c^	1.79 ± 0.48 ^d^
Dimethyl sulfide	C_2_H_6_S	62.1	760.5	0.95479	1.83 ± 0.08 ^a^	0.37 ± 0.08 ^c^	1.33 ± 0.51 ^b^	0.17 ± 0.02 ^c^
2-Methylpropanoic acid	C_4_H_8_O_2_	88.1	1503.7	1.38303	4.05 ± 0.69 ^a^	4.42 ± 0.83 ^a^	1.30 ± 0.24 ^a^	1.41 ± 0.40 ^a^
2-Pentylfuran	C_9_H_14_O	138.2	1219.4	1.25334	0.17 ± 0.14 ^a^	0.10 ± 0.07 ^ab^	0.05 ± 0.004 ^b^	0.10 ± 0.01 ^ab^
2-Ethylfuran	C_6_H_8_O	96.1	775.9	1.30114	0.03 ± 0.004 ^b^	0.03 ± 0.007 ^b^	0.03 ± 0.005 ^b^	0.10 ± 0.05 ^a^

Note: In the context of the provided information, the abbreviations represent the following; MW: Molecular Weight; RI: Retention Index; Dt: Relative Migration Time Compounds with dimer formation is denoted by the symbol “*”; The letters “^a^”, “^b^”, “^c^”, and “^d^” represent significant differences (*p* < 0.05) in volatile compounds among different regions of milk powder. Wherein, M, D, C, and H denote bovine milk powder, donkey milk powder, camel milk powder, and equine milk powder, respectively.

## Data Availability

The data used to support the findings of this study can be made available by the corresponding author upon request.

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
