# Peer review of "Discriminant Analysis of Aroma Differences between Cow Milk Powder and Special Milk Powder (Donkey, Camel, and Horse Milk Powder) in Xinjiang Based on GC-IMS and Multivariate Statistical Methods"

_foods, 2023, doi:10.3390/foods12214036_

Round 1

Reviewer 1 Report

Comments and Suggestions for Authors

All paper has been designed and expressed well. But, the discussion should be supported by other publications, and comparison of obtained data with previous studies should be added.

Please revise the title of the paper and emphasize that the evaluated milk powders belong to donkey, camel and horse milks. Also same for the abstract as well. In the abstract for the line 18: Are the 15, 14, and 23 number of the volatile compounds respectively belongs for donkey, camel and horse milk powders? Same for the line 19: Are 8, 6 and 19 number of the volatile compounds respectively belongs for donkey, camel and horse milk powders? Please indicate this more clearly. Please, reorder keywords as ‘milk powder; fingerprinting; volatile compounds; GC-IMS; PLS-DA; ROC analysis’.

These advises could be taken in consider for improving manuscript:

Page 1, Line 5: Please add comma between 1 and 2.

Page 3; Line 103: Please insert information of ‘Xinnong Enterprises’ as city name, country name.

Page 3; Line 109: Please use capital letter at the begging of each word-group begging: Description of sample, Cow milk, etc.

Page 3; Line 115: Please use ‘6 h’ instead of the ‘6 hours’.  Check and revise all manuscript for similar.

Page 3; Line 117: please insert space between: 20Pa

Page 3; Line 121: ‘20 mL’ Check and revise all manuscript for similar.

Page 3; Line Page 3; Line 123: ‘500 μL’ Check and revise all manuscript for similar.

Page 3; Line 126: ‘1 μm’ Check and revise all manuscript for similar.

Page 3; Line 132: Please insert model name, brand name, city name and county name for ‘Laboratory Analytical Viewer (LAV)’.

Page 5, Line 190: Please insert the explanation of M, D, C, H as foot-note for the Figure 1. Also the resolution is quite low.

Page 5, Line 191-193: (a) 3D Spectrogram of volatile components in different milk powders - top view; (b) 2D comparative chart of volatile components in different milk powders.

Page 6, Line 213: Please insert the explanation of M, D, C, H as foot-note for the Table 2. Please revise as C3H6O, C4H8O2, C4H8O, C4H8O, etc. Please use capital letter for the first letter of each compound. Group names of compounds (ketones, aldehydes, alcohols, esters and others) should be more visible. Please reorganize the table by considering this point. You could use uppercase letters. For retention times, all values should be same digit. Also, statistical lettering should be superscript: 23.41±4.63a.

Page 8, Line 241: ‘Li et al. [33] 's research’ and please check and revise all manuscript for similar.

Page 11, Line 358: Please reorder the figures and give each one as specificity and table, similar in Figure 2. Also, please increase the resolution of the figures.

Please add more comparison with similar previous researches

Page 13; Line 444: Please insert information of ‘National Natural Science Foundation of China’ as city name, country name.

Page 13; Line 444: Please insert information of ‘Xinnong Enterprises’ as city name, country name.

Page 13; Line 444: Please delete duplicate of ‘Enterprises’

Page 13; Line 449: Please prefer ‘support’ instead of the ‘help’.

Page 13; Line 452: Please, revise the references list according to author guidelines and insert the DOI for all references.

Comments on the Quality of English Language

Minor editing could increase the effectiveness of the paper.

Author Response

Dear Dr
Thank you for the reviewers' comments concerning our manuscript entitled "Discriminant analysis of aroma differences between cow milk powder and special milk powder in Xinjiang based on GC-IMS and multivariate statistical methods" (ID: 2673972). Those comments are all valuable and very helpful for revising and improving our paper, as well as the important guiding significance to our researches. We have studied comments carefully and have made changes which we hope meet with approval.
Here is a point-by-point response to the comments and concerns of the reviewers.
1.Comment: All paper has been designed and expressed well. But, the discussion should be supported by other publications, and comparison of obtained data with previous studies should be added.
Response: In the context of retrieved articles, the comparison of volatile components among different types of milk powder primarily encompasses breast milk powder and other varieties. The majority of these studies focus on the physicochemical properties, proteins, amino acids, and fatty acids among different milk powder types. However, some studies do involve the determination of volatile components in various milk powder types, though they lack specific comparative analysis. In accordance with your counsel, we have incorporated the results of this research and conducted a comparative analysis with our paper's data.
2.
Comment: Please revise the title of the paper and emphasize that the evaluated milk powders belong to donkey, camel and horse milks. Also same for the abstract as well. In the abstract for the line 18: Are the 15, 14, and 23 number of the volatile compounds respectively belongs for donkey, camel and horse milk powders? Same for the line 19: Are 8, 6 and 19 number of the volatile compounds respectively belongs for donkey, camel and horse milk powders? Please indicate this more clearly. Please, reorder keywords as milk powder; fingerprinting; volatile compounds; GC-IMS; PLS-DA; ROC analysis’’.
Response: In response to your advice, we have added these three types of milk powder into the title and abstract, and reorganized the keywords. The milk powder's volatile compound quantities in line 18 of the abstract represent the results obtained after PLS-DA model selection, whereas in line 19, the volatile compound quantities pertain to the results after ROC analysis validation.
3.Comment: Page 1, Line 5: Please add comma between 1 and 2.
Response:Upon your advice, we have included the commas.
4.Comment: Page 3; Line 103: Please insert information of Xinnong Enterprises as city name, country name. Page 13; Line 444: Please insert information of National Natural Science Foundation of China as city name, country name. Page 13; Line 444: Please insert information of Xinnong Enterprises as city name, country name. Page 13; Line 444: Please delete duplicate of
Enterprises. Page 13; Line 449: Please prefer support instead of the help’’. Page 3; Line 132: Please insert model name, brand name, city name and county name for Laboratory Analytical Viewer (LAV)’’.
Response: In accordance with your guidance, we have supplemented and modified the information.
5.Comment: Page 3; Line 109: Please use capital letter at the begging of each word-group begging: Description of sample, Cow milk, etc.
Response: As per your suggestion, we have capitalized the initial letters.
6.Comment: Page 3; Line 115: Please use 6 h instead of the 6 hours’’. Check and revise all manuscript for similar. Page 3; Line 117: please insert space between: 20Pa. Page 3; Line 121: 20 mL Check and revise all manuscript for similar. Page 3; Line Page 3; Line 500 μμL Check and revise all manuscript for similar. Page 3; Line 1 μμm Check and revise all manuscript for similar.
Response: Based on your counsel, we have made the necessary modifications and addressed similar issues elsewhere.
7.Comment: Page 5, Line 190: Please insert the explanation of M, D, C, H as foot-note for the Figure 1. Also the resolution is quite low.
Response: Following your advice, we have made supplementary additions and enhanced the resolution of Figure 1.
8.Comment: Page 5, Line 191-193: (a) 3D Spectrogram of volatile components in different milk powders - top view; (b) 2D comparative chart of volatile components in different milk powders.
Response: According to your advice, we have made the suggested modifications.
9.Comment: Page 6, Line 213: Please insert the explanation of M, D, C, H as foot-note for the Table 2. Please revise as C3H6O, C4H8O2, C4H8O, C4H8O, etc. Please use capital letter for the first letter of each compound. Group names of compounds (ketones, aldehydes, alcohols, esters and others) should be more visible. Please reorganize the table by considering this point. You could use uppercase letters. For retention times, all values should be same digit. Also, statistical lettering should be superscript: 23.41±±4.63a.
Response: In accordance with your guidance, we have added additional information, rectified errors, and supplemented the tables.
10.Comment: Page 8, Line 241:‘‘Li et al. [33] 's research’’and please check and revise all manuscript for similar.
Response: Following your advice, we have rectified errors and reviewed similar sections for consistency.
11.Comment: Page 11, Line 358: Please reorder the figures and give each one as specificity and table, similar in Figure 2. Also, please increase the resolution of the figures.
Response: In line with your guidance, we have restructured the formatting of the figures and improved their resolution.
12.Comment: Page 13; Line 452: Please, revise the references list according to author guidelines and insert the DOI for all references.
Response: Per your advice, we have conducted a thorough review of the references and added DOI numbers where available. However, it should be noted that for some papers, DOI numbers were not found.
Thank you and best regards.
Yours sincerely,

Reviewer 2 Report

Comments and Suggestions for Authors

ARTICLE REVIEW

Title: Discriminant analysis of aroma differences between cow milk powder and special milk powder in Xinjiang based on GC-IMS and multivariate statistical methods

Authors: Yongzhen Gou, Yaping Han, Jie Li, Xiyue Niu, Guocai Ma, Qian Xu

The work is well written.

In the introduction, the authors justified the selection of the research topic in detail.

The chapter Material and methods describes the preparation of the research material, and the research methods used in the work.

Results and discusion: the results obtained are presented in the form of 1 table and 4 figures.

The obtained data were thoroughly compared with the data of other researchers.

The conclusions are consistent with the evidence and arguments presented. The conclusions respond to the set goal of the work.

Comments:

Line 126 - a constant temperature of 60 °C for compound separation - standardize throughout the work

Author Response

Dear Dr

Thank you for the reviewers' comments concerning our manuscript entitled "Discriminant analysis of aroma differences between cow milk powder and special milk powder in Xinjiang based on GC-IMS and multivariate statistical methods" (ID: 2673972). Those comments are all valuable and very helpful for revising and improving our paper, as well as the important guiding significance to our researches. We have studied comments carefully and have made changes which we hope meet with approval.

Here is a point-by-point response to the comments and concerns of the reviewers.

Comment:Line 126 - a constant temperature of 60 °C for compound separation - standardize throughout the work.

Response: In light of your guidance, we have undertaken a revision of this section: The samples were subjected to compound separation using a WAT - Wax capillary column (30 m × 0.53 mm, 1 μm) at a constant temperature of 60°C, thereby standardizing the entire experimental procedure.

Thank you and best regards.

Yours sincerely,

Reviewer 3 Report

Comments and Suggestions for Authors

This manuscript addresses aroma compounds in four types of milk.  The authors analyzed the compounds using headspace-GC-IMS and show that this method allows them to discriminate significantly.  This manuscript provides useful information of authenticity of milk in market, however, minor modifications will be needed for publication, especially introduction section.

The authors selected some markers for discrimination using PCA and ROC. I recommend that the authors will mention origin of these compounds, material or process in discussion section.  The discussion will be useful for some readers.

Also, I recommend that the authors optimize the size of figures because some figures, especially Figure 1 and 3, so small.that it might be difficult to understand.

Author Response

Dear Dr
Thank you for the reviewers' comments concerning our manuscript entitled "Discriminant analysis of aroma differences between cow milk powder and special milk powder in Xinjiang based on GC-IMS and multivariate statistical methods" (ID: 2673972). Those comments are all valuable and very helpful for revising and improving our paper, as well as the important guiding significance to our researches. We have studied comments carefully and have made changes which we hope meet with approval.
Here is a point-by-point response to the comments and concerns of the reviewers.
1.Comment: The authors selected some markers for discrimination using PCA and ROC. I recommend that the authors will mention origin of these compounds, material or process in discussion section. The discussion will be useful for some readers.
Response: Upon your advice, we have incorporated an extensive discussion at line 380 in our paper, elucidating the underlying factors responsible for the majority of the substances generated by PLS-DA and ROC selection results, as well as the origins of variations in different milk powder formulations.
2.
Comment: Also, I recommend that the authors optimize the size of figures because some figures, especially Figure 1 and 3, so small.that it might be difficult to understand.
Response: In accordance with your counsel, we have reconfigured the layout of Figure 1 and Figure 3, while enhancing their resolution.
Thank you and best regards.
Yours sincerely,
